Darwin review 

**Cite this article:** Kokko H. 2021
The stagnation paradox: the ever-improving but (more or less) stationary population fitness. *Proc. R. Soc. B* **288**: 20212145.

evolution

female demographic dominance,
fitness measures, long-term evolution
experiment, cryptic evolution

**Author for correspondence:**
Hanna Kokko
e-mail: hanna.kokko@uzh.ch

# The stagnation paradox: the ever-improving but (more or less) stationary population fitness

## Hanna Kokko

Department of Evolutionary Biology and Environmental Studies, University of Zurich, Winterthurerstrasse 190, CH-8057 Zurich, Switzerland

(iD) HK, 0000-0002-5772-4881

Fisher's fundamental theorem states that natural selection improves mean fitness. Fitness, in turn, is often equated with population growth. This leads to an absurd prediction that life evolves to ever-faster growth rates, yet no one seriously claims generally slower population growth rates in the Triassic compared with the present day. I review here, using non-technical language, how fitness can improve yet stay constant (stagnation paradox), and why an unambiguous measure of population fitness does not exist. Subfields use different terminology for aspects of the paradox, referring to stasis, cryptic evolution or the difficulty of choosing an appropriate fitness measure; known resolutions likewise use diverse terms from environmental feedback to density dependence and 'evolutionary environmental deterioration'. The paradox vanishes when these concepts are understood, and adaptation can lead to declining reproductive output of a population when individuals can improve their fitness by exploiting conspecifics. This is particularly readily observable when males participate in a zero-sum game over paternity and population output depends more strongly on female than male fitness. Even so, the jury is still out regarding the effect of sexual conflict on population fitness. Finally, life-history theory and genetic studies of microevolutionary change could pay more attention to each other.

## 1. Introduction

Evolution refers to descent with modification over time, typically as a result of changes in gene frequencies within a population, though not all changes are reducible to competition between alleles at a prespecified and pre-existing locus [1]. Importantly, not all evolution is adaptive: if a small population accumulates so many deleterious mutations that it succumbs to mutational meltdown, evolution has certainly happened [2], but the population did not 'adapt'. Adaptive evolution, specifically, can be viewed as the ability of natural selection to produce organisms that are in some sense a better fit, compared with their ancestors, to biotic and abiotic conditions that prevail around the organism.

But what does a 'better fit' mean; what improves? The relevant concept here is population fitness, often simply phrased as 'mean fitness' [3]. Some authors equate adaptive evolution with increased population fitness [4], allowing one to be precise about the reason why an example of mutational meltdown fails to be an example of adaptation: evolution that is associated with a decline in fitness is not adaptive. This definitional choice, however, comes with downsides, as we will see below in detail. As a whole, 'population fitness' is a concept that appears straightforward in some formulations of evolution, but can behave in surprising ways when specific genotypes outcompete others.

Why the surprise? Consider the thought processes inside the head of a smart undergraduate interested in the mathematics of the situation. First, she learns

about a classic result: Fisher's fundamental theorem is stated to prove that natural selection acts to improve the mean fitness of a population [5–7]. After a brief confused moment, where she wonders how come she has also read about meltdowns and other processes of maladaptation [3], she notices that this does not yet form a contradiction. Fisher's result refers to natural selection acting on a population, and the brief confusion is resolved by categorizing maladaptive cases as examples of evolutionary change brought about processes other than natural selection. But this only paves the way to the next, more fundamental confusion.

The student now reads books and papers that equate population fitness, or mean fitness, with growth rates [5,8–11]. Now, something odd happens, when putting one and one together. Natural selection improves fitness, and high fitness implies rapid population growth. But this quickly becomes absurd, once the student reflects on the consequences over macroevolutionary timescales. The view is absurd whether she forecasts to imagine fantastically rapidly growing populations in the future, or backcasts. Did all ancestors of extant beings have really sluggish population growth rates in, say, the Triassic? Not everything alive now had evolved then yet, but ginkgoes, sharks, bacteria all existed. No one would, however, use estimated rates of evolution to predict just how slowly their populations were growing back then; but why precisely would the backcasting, should someone attempt it, be so pointless and wrong?

The problem in a nutshell: how can fitness simultaneously stagnate while it is also continually expected to improve? I will call this result the *stagnation paradox*. It is not a true paradox in the sense of being an unresolved mystery. It appears, however, that different subfields have tackled the problem somewhat independently of each other, and I hope that reviewing population fitness and its expected behaviour over time will help subfields learn from each others' progress.

## 2. Fitness in the long-term evolution experiment

I begin with a look at a famous example. The long-term evolution experiment (LTEE) is a phenomenal effort involving *Escherichia coli* populations in flasks, reaching 73 500 generations before being frozen owing to the COVID-19 pandemic. Since samples have been regularly frozen at any time, and the bacteria spring back to life once thawed, the work has led to great insights into the molecular underpinnings of ongoing adaptation. For the current context, the pertinent result is that there is support for the continual nature of fitness improvement [12].

Here, it is important to realize that an improvement in fitness does *not* imply that a newer strain will grow faster than its ancestor, when each of them exists on their own (in separate flasks). This point is frequently misunderstood, which probably contributes to the confused state of mind of our hypothetical student. For example, retrieving information on the LTEE on Wikipedia (retrieval done on 22 September 2021) states 'By 20 000 generations the populations grew approximately 70% faster than the ancestral strain', citing [13] as the source. The source itself, however, states the exact opposite when being specific about populations being tested separately: 'The evolved populations yield fewer cells than the ancestor when they are grown separately under the standard conditions' [13, p. 245]. This finding is linked to

evolved populations producing fewer but larger cells. The paradox is resolved by noting that '[e]volved cells do generally out-number the ancestors at the end of a competition assay, but not when they are grown separately' [13, p. 245].

It is instructive to go through a numerical example of what a 70% increase in fitness really means. I will do this for a hypothetical species that does not necessarily undergo the procedure of the LTEE, where 1% of the population is daily transferred to a fresh growth medium. The transfers, which have been done more than 10 000 times for the LTEE since its inception more than 30 years ago, create a permanent renewal of conditions permissive of rapid growth. I will also correct a mathematically unfortunate choice in LTEE's published results where log-transformed values appear as numerators and denominators of ratios [10]. Log transformation assigns zero or negative values to constant or declining population numbers, respectively. The nonsensical nature of log ratios becomes very clear should an ancestor be observed to keep its population size constant in a situation where its descendant, an evolved strain, is able to grow (the procedure recommended by [10] will then divide by zero), or if genotypes compete in a manner where one genotype doubles its numbers while those of its competitor are halved (the procedure then yields a nonsensical relative fitness value –1 for the winning genotype).

Correcting for this, consider a competition assay between an ancestral organism and an evolved one. For consistency with a paper that I discuss further below, I will deviate from the notation in [10] and denote the ancestral genotype A, and the evolved one A′. For brevity of notation, I will use italicized variables $A$ and $A′$ to denote population sizes of A and A′ genotypes. The populations are initiated with $A_0$ and $A′_0$ individuals. In my numerical examples below, I assume $A_0 = A′_0$ for simplicity alone; [10] gives reasons why sometimes other choices are preferable, e.g. if the decline of one genotype will make it hard to estimate its numbers at a later stage. Note that the assay has to be short enough that the competitors do not have time to evolve (significantly), i.e. we are interested in quantifying the *evolved* rather than (presently) evol*ving* differences.

Evaluating the demographic performance of the competitors in the presence of each other involves observing the population sizes $A_f$ and $A′_f$ at some time point later (f stands for 'final'). The fitness change that occurred when A evolved into A′ can be estimated as the difference between $\ln(A′_f/A′_0)$ and $\ln(A_f/A_0)$. These are the two quantities that [10] uses in a ratio, while I will quantify the fitness change as $\ln(A′_f/A′_0) - \ln(A_f/A_0)$. Here, positive values mean that evolution has led to an increase in fitness, in the sense of the evolved strain being able to outcompete the ancestral one; negative values imply a fitness decrease. This can be translated back into a change of percentages using the exp() function, thus one can speak about a 70% increase in fitness when the population evolved from A to A′, if $e^{\ln(A′_f/A′_0) - \ln(A_f/A_0)} = 1.7$.

Assume $A_0$ and $A′_0$ initially consist of a million ($10^6$) individuals each. All the following options are equivalently consistent with a 70% improvement of fitness when the ancestor A evolved into A′:

(i) both populations are able to grow in each other's presence. During the competition assay, the ancestor doubles its population ($A_f = 2 \times 10^6$) in the time that

the evolved strain grows to 3.4 million individuals ($A'_f = 3.4 \times 10^6$);

(ii) the ancestor is unable to grow but does not decline either ($A_f = A_0 = 10^6$), in the time that the evolved strain grows to 1.7 million individuals;

(iii) the ancestor declines to $A_f = 740\,740$ individuals, while the evolved strain grows to $A'_f = 1\,259\,260$ individuals. Note that these two numbers represent zero-sum competition where the total number of individuals has remained constant (two million), $A_0 + A'_0 = A_f + A'_f$, while the proportion of A′ has increased.

In every case (i) to (iii), the frequency of the evolved strain was 0.5 in the beginning of the assay, and $1.7/(1.7 + 1) = 0.6296$ at its end. That the list of examples includes a case (i) where the frequency of A declined despite A growing in absolute numbers shows that the frequency of a winning genotype (A′) can increase towards fixation even if its competitor (here, A), too, is able to grow exponentially; A′ only needs to grow faster. Even so, conditions for this to be true appear special: in the LTEE, they are regularly created through transfers to fresh medium. Biological intuition suggests that most organisms are not that fortunate with their resource supply. Indeed, it is a popular exercise in undergraduate textbooks of ecology to make a student realize how quickly unbound exponential growth becomes a physical impossibility on a finite planet. Analogously, in life-history theory, one can reflect on the concept of the Darwinian demon [14]: selection a priori favours earlier maturation, higher fecundity and improved survival, thus the optimal creature will mature immediately at birth and live forever while producing young at an infinitely high rate. The demon would fill the entire universe with its copies immediately—and because this does not happen, something about the environment must be limiting. It is therefore of interest to think about case (iii) in detail, as it imposes a strict upper limit on the number of individuals present at any time.

Case (iii) shows that genotypes can replace each other, and one can be stated to be more fit than the other, without there being any observed population growth during the replacement process. This should help resolve the confusion of the smart student working through this example. Genotype A is outcompeted and declines in both absolute numbers and in relative numbers in case (iii). However, because it is the ancestor of A′, it is biologically unlikely that it was equally unable to grow when A′ had not yet evolved. Instead, the ancestor lineage loses its ability to maintain its numbers only after the appearance of A′ as a competitor. In the language of Fisher, the environment that A experiences has deteriorated [5].

Fisher's writings offer an intriguing mix of crystal clear insight and a severe case of the 'curse of knowledge', the phenomenon that once one has understood something well, it is hard to convey the message to others because one no longer remembers what it was like to not know it. His choice of the phrase 'environmental deterioration' [5] is a good example of this curse. I have witnessed reviewers complain that one shouldn't assume environments generally decline from the perspective of an organism, as conditions might equally often improve (at least in the absence of environmental problems of anthropogenic origin). When hearing 'environmental deterioration' for the first time, the fact that a conspecific representing an alternative genotype may be snatching resources that an individual of the focal genotype could have consumed is definitely not the first mental image that springs to mind. Yet this image is key to understanding the stagnation paradox [15]. A given genotype will experience 'environmental improvement' if competitors spread that are inferior, and 'deterioration' if they are superior; but because inferior competitors are not likely to spread, while superior ones do, deterioration is expected to predominate.

It is also instructive to realize that the thought process involving (iii) can be performed for longer than the initial assay. A 70% fitness difference in a finite population that is continually kept at $A + A' = 2$ million individuals implies that the ancestral strain is bound to disappear before long (this result requires finiteness of populations: in a population of infinite size one could still have a vanishingly small frequency of A present at any time, declining exponentially, with zero as its asymptote). If one keeps assuming that the total population is always 2 million, one has, after the extinction of A, reached a state where there is no way for A′ to show its 70% higher fitness. When A′ is fixed, on average every A′ individual only replaced itself, instead of producing 1.7 copies. An observer arriving on the scene now would state that mean fitness is 1 (on average), not 1.7. In other words: while A′ was clearly fitter than A, limited ecological resources make it impossible for A′ to showcase its improved population fitness in any demographic sense, after A′ has outcompeted A.

## 3. Serial rediscovery?

Kay et al. [16] recently argued that confusing terminology, modelling choices and differences in 'schools of thinking' have fuelled unnecessary fights about the role of relatedness in the evolution of cooperation, using the phrase 'serial rediscovery' in their title to describe the process. The stagnating nature of fitness has similarly been rediscovered independently in different fields, though here with a difference that there is no heated fight about the phenomenon or how to explain it. Instead, different authors simply appear to some extent unaware of some parts of the previous literature. I will here trace a few of the important contributions.

The precise meaning of Fisher's 'environmental deterioration' was the focus of two important papers in the early 1990s [15,17]. Cooke et al. [15] built a small but insightful model of clutch size evolution, and Frank & Slatkin [17] reflected in more detail on how Fisher's results relate to the evolution of fitness in general. I will present the argument with a brief recap of points made in [17] using the notation of A and A′ as introduced above. They noted that Fisher was fully aware that evolution causes a change in traits, and that the population-wide values of these traits also have to be considered an important part of an organism's environment. Evolution makes the environment, as it is 'felt' by an organism, change from one containing predominantly A competitors to one where one is surrounded by A′ types.

Now, consider fitness of the ancestral population ($\bar{W}$) as well as that of the evolved one ($\bar{W}'$). When the environment consists of A individuals, then in Frank & Slatkin's terminology this is the environment E; later, when all individuals are A′, the environment has become E′. The important points are that the values of $\bar{W}$ and $\bar{W}'$ are specific to the environment they are measured in, and that Fisher's fundamental theorem does not predict anything specific about the value of

$\bar{W}'|E'-\bar{W}|E$. If it did, then we would indeed predict that populations grow now generally faster than in the Triassic. Instead, what Fisher predicts to be positive is only term 1 of the decomposition:

$$\bar{W}'|E' - \bar{W}|E = \underbrace{\bar{W}'|E-\bar{W}|E}_{\text{term 1}} + \underbrace{\bar{W}'|E'-\bar{W}'|E}_{\text{term 2}}. \qquad (3.1)$$

This first term states that the improvement relates to competition in environments that were the result of the ancestral population and its ecological effects. In other words: if measured in the same (ancestral) environment E, the evolved genotype outcompetes the ancestral one. Given that the *whole* expression (3.1) is not positive, at least not consistently over large swathes of time (to avoid incorrect statements about the Triassic), and the first term is positive as a result of natural selection, it is a reasonable expectation that the second term is typically negative. Its negativity shows that the evolved genotype performs less well in the environment that it itself created than what it experienced when beginning to invade, hence the phrase 'environmental deterioration'.

The way one arrives at the positive first term also captures the essence of the competition assay, though with a slightly altered experimental design: an idealized experiment would measure how well an evolved strain would do in an environment where almost every individual is still the ancestral strain. This is also the gist of any mathematical approach that uses evolutionary invasion analysis [18,19]. In real competition assays, there are practical reasons why the invading, evolved, genotype is not introduced as a very rare strain. Very rare invaders can be lost owing to drift and may generally be hard to detect; therefore, the favoured approach is a more even mix of ancestral and evolved genotypes, and I phrased my examples (i)…(iii) to reflect this.

## 4. The life-history angle

My claim of there being a 'serial rediscovery' is most evident in an important paper by Mylius & Diekmann [20], that is firmly rooted in life-history theory. Life-history theory tackles head-on the question of what precisely prevents Darwinian demons from existing, with insights such as parents of too large broods experiencing difficulty surviving themselves to breed again [21–23]. If the trade-off structure of a given situation is known, one will, in principle, be able to find out the phenotypic value of a trait, e.g. clutch size, that will maximize fitness. Before [20] was published, there was a significant stumbling block: how to practically quantify fitness itself (review: [24])? If we know a given scheduling of reproduction and survival probabilites involved in a specific life cycle, what is the value of fitness that we can summarize it with?

One possibility is to count the expected number of offspring produced in a lifetime (denoted $R_0$, or LRS for 'lifetime reproductive success'). However, this was already known to be potentially incorrect because, in a growing population, early reproduction is favoured [24]. This is not only because a 'plan' to produce offspring later might never materialize if the parent is dead. Instead, or additionally, offspring are predicted to themselves contribute more to future generations if their parent manages to place them into a population earlier (one way to see it is that the early population is smaller than its later version, thus one offspring placed into it earlier forms a larger proportion of the gene pool). The correct fitness measure then would appear to be $r$, the population growth rate of a lineage that uses a specific life-history strategy. However, as we have seen from the example (iii) above, adaptive evolution is absolutely compatible with no change in population numbers at all (zero growth), and unless one is observing *E. coli* in the LTEE, one can hardly expect indefinite growth to prevail; so how can we use $r$ as a fitness measure if both 'before' and 'after' evolutionary change we simply observe $r = 0$?

Mylius & Diekmann [20] solved this puzzle elegantly, showing that an informed (and uniquely correct) decision of the fitness measure can only be made if one makes it explicit how the population is regulated. If the evolution of, say, clutch size leads to increasing densities, this then deteriorates the environment for each population member so that the expected lifetime reproductive success is multiplied by a factor of less than 1 and [20] proves that in this case it remains fully valid to use $R_0$ as the fitness measure. Heuristically, earliness is no longer favoured, given that offspring placed into the population earlier do not form a larger subset of a population that does not change its size. However, there are also situations where $r$ remains the correct fitness measure, and others where neither is. Solutions can be found for these 'other' cases too, but not without specifying, explicitly, how environmental feedback, in the form of density dependence, operates to regulate the population.

I mentioned clutch size above, but Mylius & Diekmann [20] actually made extensive use of age at maturity, not clutch size, as their most elaborate example of a life-history trait. Their talk about 'environmental feedback' also made them narrowly miss using 'environmental deterioration' as a phrase, even though they came close: they speak of moving from a 'virgin environment' to an 'environmental condition' that was, where appropriate, stated to be worse than the virgin one. It is tempting to speculate whether they would have discovered the clear links between their findings and those of Cooke *et al.* [15] and Frank & Slatkin [17] had they considered clutch sizes as an example ([20] does not cite [15,17] or any of Fisher's work for that matter). Perhaps it is too optimistic to assume that an overlap of the chosen trait would have been sufficient to spark cross-fertilization of these fields: quantitative genetics, with its interest in predicting ongoing microevolution, and optimization approaches to life-history theory, which has the goal of trying to predict the precise location of the ultimate fitness peak, are to this day not cross-referencing each other very frequently. Attempts to point out equivalences and relationships between them exist [25–33] but for some reason do not appear to build much information flow between the mainstream currents of each field. As is, the insights in [20] have had a major impact on life-history theory, less so on the continued discussion that has Fisher [5] and Cooke *et al.* [15] at its origin.

In the early 2000s, Merilä *et al.* [34] continued the discussion in quantitative genetics with a focus not on fitness *per se* but on particular traits that covary with fitness. They provided data for fledgling condition as a trait that is under positive selection but fails to improve phenotypically and also coined the name 'cryptic evolution' for the case where breeding values for the trait improve but the phenotype does not. In another paper [35], they discussed numerous examples and potential reasons (including temporal fluctuations and statistical sampling problems) for phenotypic

stasis. When discussing environmental change as a potential reason, they placed more emphasis on types of change that are not causally related to the change from E to E′ as described above. This makes them appear to not have fully grasped the near inevitablity of environmental deterioration, in the sense of the positive and negative terms in equation (3.1). Thus, despite the importance of their contributions, they [35] do not link their findings to either Frank & Slatkin [17] or Mylius & Diekmann [20]. Cooke *et al.* [15] is cited, however, though the writing might not make it clear that environmental deterioration happens even if observed densities do not rise (like in case (iii) above). Fisher's classic work [5] is cited, but this book-length work is discussed for a different point in [35]: that natural selection may act predominantly on the non-heritable component of a phenotype, which then explains the lack of a response to selection.

A decade later, Hadfield *et al.* [36] provided a necessary fix. They further clarified and unified previous work, casting Cooke *et al.*'s model in terms of quantitative genetics and reinstating the importance of Fisher's concept of deterioration. They suggest, very sensibly, calling it 'evolutionary environmental deterioration' to help distinguish it from numerous other processes that can also impact trait evolution and the resulting phenotypes; the effects are, after all, not mutually exclusive. However, the links to life-history work and the wide-ranging importance of [20] for eco-evolutionary modelling up to this day (present-day examples: [37–40]) were not noted by [36], and the fields continue to being quite separate to this day.

## 5. It is not all about food: individual success at the expense of conspecifics

Population fitness, or 'mean fitness of a population', perhaps surprisingly does not have a unique definition. By this I mean that there are many contrasts among the $\bar{W}'|E'$, $\bar{W}|E$, $\bar{W}|E'$ and $\bar{W}'|E$ that one could conceivably mean when stating 'population fitness has improved'—and what about non-equilibrium situations brought about environmental change other than 'evolutionary environmental deterioration' *sensu* [36]? Non-equilibrium situations yield, at least temporarily, an environment that deviates from both E and E′. Intuitively, population fitness should be reflected in the ability of populations to bounce back from unfortunate situations such as harvesting or predators killing a part of the population, or drought suddenly removing much of the food. Here, one might assume Fisher's fundamental theorem to predict that those populations that have experienced adaptive evolution (for longer or faster) before the adverse event will have improved performance when growing back to carrying capacity. When populations are perturbed away from density-dependent equilibria, they might show fitness differences akin to my case (i) above.

While it is true that fitness differences can manifest themselves in faster growth rates, it is once again a naïve expectation that natural selection will always favour traits that enable such responses. Of course, it is tempting to imagine that an organism which has evolved superior foraging skills, or is physiologically better able to convert food into offspring, will form populations that are more resilient during or after adverse events. This hypothesis is probably also often true. The reason that not all cases are like this, however, is that not all traits are similar to foraging efficiency, and

what evolves is then not equivalent to what maximizes fitness [41,42].

To be precise about the hypothesis we are investigating, let us state the contents of the generalized hypothesis in two parts:

— while $\bar{W}'|E'$ may be no larger than $\bar{W}|E$ (e.g. for both it can be true that $R_0 = 1$, equivalently phrased as $r = 0$, such that each individual on average replaces itself from one generation to the next), …
— … when the population is in an environment that places it below carrying capacity, a population of evolved A′ individuals will show superior growth than their ancestors A would have done.

Why is it possible to find cases against the hypothesis? Even if selection acts on a trait and there is a response to selection, it is possible that the evolved trait values, once spread to be expressed by all (or most) population members, fail to show a better fit to the environment or better demographic performance for any of the relevant vital rates. (Conversely put, the population growth rate produced by a trait or a strategy *when fixed in a population* is not an infallible tool for predicting which trait value will outcompete its alternatives.) Potential examples are not at all hard to find. Trees compete for sunlight and attempt to outshade each other, but when each tree consequently invests in woody growth, the entire forest must spend energy in stem forming and—assuming time or energy trade-offs—will be slower at converting sunlight into seeds than a low mat of vegetation would have been able to [43]. Every individual has to invest in outcompeting others, but the population as a whole is negligibly closer to the light source (the number of photons arriving in the area is still the same). This is why in agriculture, externally imposed group selection to create shorter crops has improved yields [44]. Similarly, migrant birds may try to outpace each other in a quest to occupy the best breeding positions, even though a more relaxed arrival schedule would presumably allow them to arrive in a better condition and conceivably boost population growth and/or provide resilience in case of adverse early-spring weather; average quality ranks of territories gained would be equally good in a non-rushed arrival scenario, thus the 'rush' is pointless from a group perspective [45].

Note that it is possible to consider an alternative form of this hypothesis, where the evolved A′ is hypothesized to reach higher population densities or total size than A did, instead of focusing on performance under non-equilibrium conditions. There are indeed conditions where natural selection leads to maximizing population size ([46, p. 168]), and large population sizes might then offer resilience against extinction or exploitation [47,48]. Yet, this alternative would not change the conclusion: it is possible that contests among conspecifics select for traits that reduce population size relative to what alternative trait values would dynamically predict [49].

The general conclusion is that if there is a way for a genotype to succeed disproportionately by exploiting conspecifics, then relative fitness differences take precedence over the maintenance of population fitness. Population fitness may be particularly prone to behaving in unexpected and diverse ways if one subset of the population participates in zero-sum games causing positive or negative, direct or

indirect effects on the reproductive success of another subset of individuals. In short: males may compete for females in ways that impact female fitness.

Equation (3.1) above did not separate between females and males at all, and this of course is not relevant for a case like the LTEE: there are no male or female *E. coli*. In anisogamous species, however, females tend to be demographically dominant [50], meaning that their performance is a stronger determinant of population growth than the number, or fitness, of males. One way to phrase this is that selection is softer on males than on females [51], and it is well known that soft selection can maintain population fitness at demographically adequate levels even if a straightforward computation of mutational load would predict dismal performance [52,53]. Classic theory on load and soft selection, however, does not differentiate between males and females, and because reproduction often involves direct behavioural contact between contestants or potential mates, very diverse outcomes are possible.

In particular, sexual selection offers such good examples of population-level harm that it has been popularized in accounts of behavioural economics as counterexamples for naïve expectations that free markets will automatically promote a population-wide good [54]. Using antlers of elk as an example, Frank ([54]; a different Frank from that of [17]) points out that greater investment in trait exaggeration does not make males as a group better fit (alternatively put, the mean mating success of males will not improve if all of them fight better). A small-antlered population would also find and fertilize females, and avoid the costs of growing and maintaining antlers.

While it is easy to agree with the general gist of the statement, and generally expect that sexual or social traits can evolve to exploit conspecifics even if they are detrimental to group or population fitness [55–58], the question of the net effect is ultimately an empirical one. In bovids, large horns appear to increase extinction risk [59], and for sexual selection as a whole, the jury is still out [42,59–65]. A mix of results, from negative to positive, is not wholly unexpected in a situation where theory shows sexual selection to have the potential to elevate female fitness. Beneficial effects at the genetic level arise because selection against deleterious genetic variants operates more strongly on sexually selected males than females [66]. In such a setting, the gene pool is purged without causing a large demographic cost on females, and such processes may be particularly important when both sexes are far away from the current fitness peak [67,68]. Yet, simultaneously, any such benefit has to be pitted against the numerous plausible processes where the male response to selection depresses female fitness either directly (when males harass females, evolve seminal fluids that are detrimental to females, or simply grow larger than them and deplete local resources) or indirectly, via intralocus sexual conflict where females express genes that improve fitness in males but not in females.

There are numerous examinations of how sexual selection and sexual conflict relate to growth or stability, resilience against extinction via inbreeding, or other proxies of population fitness (review: [69], newer work: [59–65,70–74]). To make sense of the diverse mess, systematic approaches appear welcome. Recently, Cally *et al*. [75] took on such a task and performed a meta-analytic summary of a specific subset of approaches: those where experimental evolution [76] has been used to evaluate whether populations evolving under sexual selection (typically brought about by polygyny) led to improved fitness. Because of all the possibilities listed above, it is hardly surprising that effect sizes ranged from negative to positive. Intriguingly, the more direct the link between the choice of measure and population fitness, the *less* strong the evidence that sexual selection typically has a positive effect.

The study by Cally *et al*. [75] is also thought provoking in its statements that male fitness can change in ways that are not directly linked to typical responses in females. They state that fitness benefits (of sexual selection) to males were weakened in stressful conditions, while the opposite was true for females. While the data indeed show this pattern, the take-home message is less clear. To see why, consider mean fitness for males, whenever they participate in a zero-sum game over paternity. Males can, as a group, only sire as much progeny as females (in the current social and sexual setting) are able to produce and have fertilized. Individual males, of course, can become fitter, but reproductive success should then occur at the expense of others, and whenever one individual's win is another's loss, mean fitness, in terms of realized reproductive success of an average male, ought not to change. The fact that male fitness did change for many measures reported in [75], and that responses were not simply a function of mean reproductive success of females, is only explicable if the studies were reporting proxies of fitness that do not particularly directly relate to population fitness and are, consequently, probably unreliable predictors of resilience to environmental change. (This of course does not prevent them from being interesting in their own right, to explain what evolves in males.) It therefore remains a hopefully active field to understand when exactly trait evolution elevates population fitness in any meaningful demographic manner.

## 6. One more angle, and one more definitional minefield: indirect genetic effects and adaptation

In the context of social evolution, the mathematical side of the stagnation paradox has recently been worked on (arguably 'rediscovered') by Fisher & McAdam [4]. They examine the by now familiar fact that overall performance of groups or populations can evolve in a detrimental direction [56] using the language of indirect genetic effects (see also [77]). Now the environment specifically refers to the social environment, which thus could be seen to be a special case of Hadfield *et al*.'s [36] evolutionary environmental deterioration (though [4] appear not to have been inspired by [36], evidenced by the lack of citation). The indirect genetic effect approach successfully reconciles Fisher's fundamental theorem with the Price equation, by pointing out that the transmission bias term in the latter is typically not zero if others' success has a negative effect on a focal individual's success.

This is the same effect as the numerous examples where adaptation to outcompete conspecifics leads to situations characterizable as the tragedy of the commons [56]. Fisher & McAdam [4], however, go further and propose that the word 'adaptation' be restricted to cases where evolutionary change improves population fitness (which they phrase as 'mean fitness', with their examples clearly showing that this is a population-wide measure of reproductive

output). Above, I have already commented on the confusion that can follow from different interpretations of the word 'environment' in the context of population fitness. The interpretation of the word 'adaptation' in [4] is unusual. Because they restrict adaptation to cases where population fitness (in their terminology: mean fitness) improves, it forces them to explicitly classify armament and ornament evolution as examples of non-adaptive evolution. I believe there are two reasons why a population fitness-based definition of adaptation is too restrictive.

The first reason is its potential to confuse, as it conflicts with widespread usage of adaptation to mean a process of organisms evolving traits that appear designed to achieve a particular goal [78]. Although careful study is in each purported case required to establish whether a particular goal really is the answer, in the case of an armament (e.g. antlers of male deer), the answer is fairly clear. It is very likely to approximate to 'to establish one's status within a dominance hierarchy to achieve matings'. Likewise, for ornaments, the answer is likely to be along the lines 'to impress a female enough so that she solicits a mating'. It would be unfortunate to classify evolutionary processes that explain traits with an apparent purpose [79] in the same non-adaptive bag as mutational meltdowns.

The second reason to avoid defining adaptation via population fitness improvements is a pragmatic one: as I hope to have shown above, any value of population fitness is very sensitive to the ecological context that it is measured in. In any real population (outside, perhaps, LTEE's flasks), it is easier to observe $\bar{W}'|E'$ and $\bar{W}|E$, the fitness of individuals in an environment they themselves create, than it is to measure $\bar{W}'|E$ or $\bar{W}|E'$ (though the latter two are not impossible if such situations can be created experimentally). Therefore, should one make decisions about adaptation based on the sign of $\bar{W}'-\bar{W}$ measured in the same environment, the task is quite an arduous one as it requires creating a suitable environment for measurement, and observational field data will hardly ever be suitable. If, on the other hand, one decides that $\bar{W}'|E'-\bar{W}|E$ is a criterion, then evolutionary environmental deterioration *sensu* [36], or ecological feedback *sensu* [20], interfere immediately with the reasoning. The results [20,36] remind us that $\bar{W}'|E'-\bar{W}|E$ is often, for purely ecological reasons, near zero and would fluctuate between 'adaptation' and 'maladaptation' in a stochastic manner, should we adopt the sign of the difference as a criterion to determine between these two important concepts.

## 7. A brief conclusion

I will end with a metaphor. Evolutionary theory has an analogy in its own object of study. The tree of life has clear origins in unicellular life, just as all evolutionary thinking can be traced back to the writings of Darwin and a few of his predecessors and contemporaries. Later developments occur along with many branches, and sometimes there is convergent evolution where a similar solution to a problem is found independently—and, perhaps, more painstakingly than if flow of information was unrestricted. Given that 'all paths to fitness lead through demography' [80], and demography is generally modulated by density dependence and biological interactions between conspecifics, it would be wonderful if demographic studies, and the phenomenon variously termed evolutionary environmental deterioration or eco-evolutionary feedback, were firmly part of the mindset of anyone asking microevolutionary questions—and why not macroevolutionary, too.

**Data accessibility.** This article has no additional data.

**Competing interests.** I declare I have no competing interests.

**Funding.** I received no funding for this study.

**Acknowledgements.** I thank Innes Cuthill for inviting me to write this piece, two anonymous reviewers for their suggestions (sorry that I did not adopt all suggestions), and the Kokkonuts journal club for countless afternoons of inspiration and fun. I would also like to thank an anonymous reviewer of a wholly different manuscript decades ago, who prevented me from writing something silly and encouraged me to read Frank & Slatkin's paper instead.

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
