## [Peer Review File · Proceedings of the Royal Society B: Biological Sciences]

Review History

RSPB-2021-2145.R0 (Original submission)

Review form: Reviewer 1

Recommendation

Accept as is

Scientific importance: Is the manuscript an original and important contribution to its field?

Excellent

General interest: Is the paper of sufficient general interest?

Excellent

Quality of the paper: Is the overall quality of the paper suitable?

Excellent

Is the length of the paper justified?

Yes

Should the paper be seen by a specialist statistical reviewer?

No

Do you have any concerns about statistical analyses in this paper? If so, please specify them explicitly in your report.

No

It is a condition of publication that authors make their supporting data, code and materials available - either as supplementary material or hosted in an external repository. Please rate, if applicable, the supporting data on the following criteria.

Is it accessible?

N/A

Is it clear?

N/A

Is it adequate?

N/A

Do you have any ethical concerns with this paper?

No

Comments to the Author

This manuscript gives a great overview of the links between adaptation, competition, population fitness and the fundamental theorem of natural selection. Some of these points are incredibly important for lots of theoretical and empirical questions, and are generally presented in some form in undergraduate teaching, yet not fully grasped by a large portion of empiricists.

Hammering them again, in a very clear format, as this manuscript does has the potential to have a broad impact on the quality of ecology and evolution research. At the very least this manuscript should prevent some researchers from having to rediscover the same 91 years old results.

I think I agree with every statement, and I really like the casual and clear writing style. Arguably one could expand this manuscript to be a few hundred pages long to include all the relevant research and nuanced points of view, but the current treatment appears sufficient and very helpful as it is.

typos?

L. 128. "caster" -> "faster" ? (or did I miss a baking metaphor?)

L.422 "is be" -> "is" ?

Review form: Reviewer 2

Recommendation

Accept with minor revision (please list in comments)

Scientific importance: Is the manuscript an original and important contribution to its field?

Excellent

General interest: Is the paper of sufficient general interest?

Excellent

Quality of the paper: Is the overall quality of the paper suitable?

Good

Is the length of the paper justified?

Yes

Should the paper be seen by a specialist statistical reviewer?

No

Do you have any concerns about statistical analyses in this paper? If so, please specify them explicitly in your report.

No

It is a condition of publication that authors make their supporting data, code and materials available - either as supplementary material or hosted in an external repository. Please rate, if applicable, the supporting data on the following criteria.

Is it accessible?

N/A

Is it clear?

N/A

Is it adequate?

N/A

Do you have any ethical concerns with this paper?

Yes

Comments to the Author

In this Darwin Review, Hanna Kokko discusses the so-called stagnation paradox: how can mean fitness improve by natural selection and yet stay constant? I really enjoyed reading the paper. The text is readable and insightful. I think the argument is important, the author does a nice job laying it out.

My comments are minor and focus on the few instances where I found the presentation to be less lucid.

ABSTRACT

Lines 9-10, "Fitness, in turn, is equated with population growth" change to "can be" or "is often"?

Lines 21-22, "Even so, the jury is still out regarding the effect of sexual conflict on population fitness". This sentence does not come naturally from the previous one

INTRODUCTION

Lines 27-29, This a long opening sentence, especially as it is written in a passive voice.

Line 50, I think it would be good to clarify who is equating mean fitness with growth rate

Lines 60-64, Would it be worth it to explain link (or lack thereof) between the stagnation paradox and the Eldredge and Gould's "stasis is data" slogan somewhere in the Introduction?

FITNESS IN THE LONG-TERM EVOLUTION EXPERIMENT (LTEE)

Line 89 Delete "of"

Line 97, I found the reference to [10]'s use of B and A to be distracting.

Lines 115-123, I think these three scenarios would make for a nice figure

SERIAL REDISCOVERY

Line 191, I think it be good to spell out W' | E' - W | E verbally

A BRIEF CONCLUSION

Lines 435-444, I like the analogy of the history of evolutionary theory as a tree with multiple branches. I feel, however, that it feels more like the premise rather the conclusion of the paper. Move to Introduction?

Decision letter (RSPB-2021-2145.R0)

19-Oct-2021

Dear Hanna

I am pleased to inform you that your manuscript RSPB-2021-2145 entitled "The stagnation paradox: the ever-improving but (more or less) stationary population fitness" has been accepted for publication in Proceedings B... well of course it has!

As you will see, both referees are very enthusiastic about your review, as am I, but they have made some useful suggestions to make it EVEN better. Therefore, I invite you to respond to the referees' comments and revise your manuscript. Because the schedule for publication is very tight, it is a condition of publication that you submit the revised version of your manuscript within 7 days. If you do not think you will be able to meet this date please let us know.

Very best wishes, and thanks again for an outstanding review,

Innes
 Professor Innes Cuthill
 Reviews Editor, Proceedings B
 mailto: proceedingsb@royalsociety.org

Reviewer(s)' Comments to Author:

Referee: 1

Comments to the Author(s)

This manuscript gives a great overview of the links between adaptation, competition, population fitness and the fundamental theorem of natural selection. Some of these points are incredibly important for lots of theoretical and empirical questions, and are generally presented in some

form in undergraduate teaching, yet not fully grasped by a large portion of empiricists. Hammering them again, in a very clear format, as this manuscript does has the potential to have a broad impact on the quality of ecology and evolution research. At the very least this manuscript should prevent some researchers from having to rediscover the same 91 years old results. I think I agree with every statement, and I really like the casual and clear writing style. Arguably one could expand this manuscript to be a few hundred pages long to include all the relevant research and nuanced points of view, but the current treatment appears sufficient and very helpful as it is.

typos?

L. 128. "caster" -> "faster" ? (or did I miss a baking metaphor?)

L.422 "is be" -> "is" ?

Referee: 2

Comments to the Author(s)

In this Darwin Review, Hanna Kokko discusses the so-called stagnation paradox: how can mean fitness improve by natural selection and yet stay constant? I really enjoyed reading the paper. The text is readable and insightful. I think the argument is important, the author does a nice job laying it out.

My comments are minor and focus on the few instances where I found the presentation to be less lucid.

ABSTRACT

Lines 9-10, "Fitness, in turn, is equated with population growth" change to "can be" or "is often"?

Lines 21-22, "Even so, the jury is still out regarding the effect of sexual conflict on population fitness". This sentence does not come naturally from the previous one

INTRODUCTION

Lines 27-29, This a long opening sentence, especially as it is written in a passive voice.

Line 50, I think it would be good to clarify who is equating mean fitness with growth rate

Lines 60-64, Would it be worth it to explain link (or lack thereof) between the stagnation paradox and the Eldredge and Gould's "stasis is data" slogan somewhere in the Introduction?

FITNESS IN THE LONG-TERM EVOLUTION EXPERIMENT (LTEE)

Line 89 Delete "of"

Line 97, I found the reference to [10]'s use of B and A to be distracting.

Lines 115-123, I think these three scenarios would make for a nice figure

SERIAL REDISCOVERY

Line 191, I think it be good to spell out $W' | E' - W | E$ verbally

A BRIEF CONCLUSION

Lines 435-444, I like the analogy of the history of evolutionary theory as a tree with multiple branches. I feel, however, that it feels more like the premise rather the conclusion of the paper. Move to Introduction?

Author's Response to Decision Letter for (RSPB-2021-2145.R0)

See Appendix A.

Decision letter (RSPB-2021-2145.R1)

22-Oct-2021

Dear Dr Kokko

I am pleased to inform you that your manuscript entitled "The stagnation paradox: the ever-improving but (more or less) stationary population fitness" has been accepted for publication in Proceedings B.

Data Accessibility section

Open Access

Paper charges

You are allowed to post any version of your manuscript on a personal website, repository or preprint server. However, the work remains under media embargo and you should not discuss it

with the press until the date of publication. Please visit <https://royalsociety.org/journals/ethics-policies/media-embargo> for more information.

Sincerely,
Proceedings B
<mailto:proceedingsb@royalsociety.org>

Appendix A

Dear Innes

Thank you, that was quite a set of reviews – rarely are they so positive!

I'd like to apologize for taking up so few of reviewer 2's suggestions, many of them made sense but then were not so easy to implement after all (see below). Below are all my responses:

L. 128. "caster" -> "faster" ? (or did I miss a baking metaphor?)

Indeed, no baking metaphor was intended...

L.422 "is be" -> "is" ?

Thanks for spotting this typo! Corrected.

Referee: 2

ABSTRACT

Lines 9-10, "Fitness, in turn, is equated with population growth" change to "can be" or "is often"?

'often' added.

Lines 21-22, "Even so, the jury is still out regarding the effect of sexual conflict on population fitness". This sentence does not come naturally from the previous one

I reflected on this and I actually think it does come naturally – though it perhaps requires thinking about the preceding 2 sentences not just 1. (The gist is: Adaptation can lead to declining population fitness; particularly so when we think about males; but males aren't always a negative force.) I tried to think how to make this clearer, but in the end gave up.

INTRODUCTION

Lines 27-29, This a long opening sentence, especially as it is written in a passive voice.

I shortened it somewhat ('current views acknowledging' removed).

Line 50, I think it would be good to clarify who is equating mean fitness with growth rate

I now cite a few papers here (there are plenty to choose from, these are just examples).

Lines 60-64, Would it be worth it to explain link (or lack thereof) between the stagnation paradox and the Eldredge and Gould's "stasis is data" slogan somewhere in the Introduction?

I spent some time thinking about it, but I believe this macroevolutionary angle would distract more than it would illuminate in that context. I end the last paragraph of the main text with a mention of macroevolution, and perhaps it's best to stop there.

FITNESS IN THE LONG-TERM EVOLUTION EXPERIMENT (LTEE)

Line 89 Delete "of"

Done.

Line 97, I found the reference to [10]'s use of B and A to be distracting.

Indeed, I reformulated.

Lines 115-123, I think these three scenarios would make for a nice figure

This is somewhat true – and I even drew them, but then ended up thinking that they failed to really capture the essence of the argument, which is the way that A' reverts back to zero growth (fitness 1). It was difficult to graph that without lots of explanation, and in the end I decided that it's better explained in the text this time, than in figures.

SERIAL REDISCOVERY

Line 191, I think it be good to spell out W|E' – W|E verbally

I agree, yet when I tried, I ended up simply replicating the Triassic argument when I tried to keep it simple and intuitive – and that already comes in the next sentence. Alternatively, one could say these to be population fitnesses at two different

timepoints, but then I would contradict myself with elsewhere in the MS because I say that many different contrasts could be made! I wrestled with this a little and then ended up not changing the current structure.

A BRIEF CONCLUSION

Lines 435-444, I like the analogy of the history of evolutionary theory as a tree with multiple branches. I feel, however, that it feels more like the premise rather the conclusion of the paper. Move to Introduction?

Again, I considered doing this but then the paper ended too abruptly with a criticism of one particular paper. I did not really want to extend it artificially, so in the end I left the ending as is.